# Multi-kinase control of environmental stress responsive transcription

**Kieran Mace[1], Joanna Krakowiak[2], Hana El-Samad[1]\*, David Pincus[2,3]\***

**1** Department of Biochemistry and Biophysics, California Institute for Quantitative Biosciences, University of California San Francisco, San Francisco, California, United States of America, **2** Whitehead Institute for Biomedical Research, Cambridge, Massachusetts, United States of America, **3** Department of Molecular Genetics and Cell Biology, Center for Physics of Evolving Systems, University of Chicago, Chicago, Illinois, United States of America

\* hana.el-samad@ucsf.edu (HES); pincus@uchicago.edu (DP)

**Data Availability Statement:** Sequencing data are deposited on GEO with accession number: GSE115556.

**Funding:** This work was supported by NIGMS grant RO1 GM119033 to H.E.-S, NIH Early Independence Award DP5 OD017941 to D.P., and

## Abstract

Cells respond to changes in environmental conditions by activating signal transduction pathways and gene expression programs. Here we present a dataset to explore the relationship between environmental stresses, kinases, and global gene expression in yeast. We subjected 28 drug-sensitive kinase mutants to 10 environmental conditions in the presence of inhibitor and performed mRNA deep sequencing. With these data, we reconstructed canonical stress pathways and identified examples of crosstalk among pathways. The data also implicated numerous kinases in novel environment-specific roles. However, rather than regulating dedicated sets of target genes, individual kinases tuned the magnitude of induction of the environmental stress response (ESR)–a gene expression signature shared across the set of perturbations–in environment-specific ways. This suggests that the ESR integrates inputs from multiple sensory kinases to modulate gene expression and growth control. As an example, we provide experimental evidence that the high osmolarity glycerol pathway is an upstream negative regulator of protein kinase A, a known inhibitor of the ESR. These results elaborate the central axis of cellular stress response signaling.

## Introduction

Natural selection confers fitness to organisms that are able to adapt to environmental fluctuations. Changes in temperature, osmolarity and nutrient availability are recurrent stresses, and cells have evolved mechanisms to specifically sense and react to these and a variety of other environmental and internal perturbations. Such adaptive processes, collectively known as stress responses, have been extensively characterized at the transcriptional level in the model eukaryote *Saccharomyces cerevisiae* (budding yeast) [1–5]. Classical genetic and biochemical studies defined dedicated signaling pathways that sense and transmit several stress cues, including hyperosmotic shock, glucose starvation, and endoplasmic reticulum (ER) stress [6–9]. More recently, genome-wide genetic interaction studies have comprehensively quantified the effects of gene deletions on several stress response pathways including the ER unfolded protein response (UPR) and cytosolic heat shock response–not only identifying core signaling

an International Student fellowship from HHMI to K.M. The funders had no role in study design, data collection and analysis, decision to publish, or preparation of the manuscript.

**Competing interests:** The authors have declared that no competing interests exist.

components but also modifiers of the responses [10, 11]. Most stress signaling pathways contain kinases that relay extracellular and subcellular information to transcription factors that control gene expression in the nucleus. However, stresses such as heat shock and oxidative damage–well characterized transcriptionally–have no kinase networks associated exclusively with them.

There are 129 kinases encoded in the *S. cerevisiae* genome, and high-throughput investigations have defined the protein-protein and genetic interactions among the members of the yeast "kinome" [12]. These analyses established the functional organization of the global kinase network and revealed mechanisms of redundancy and crosstalk in cell cycle regulation and developmental pathways [12–14]. However, the wiring of the kinome is not static. Under hyperosmotic stress conditions, genetic interactions among kinases are reconfigured, suggesting a plasticity to the underlying biochemical interactions [15]. Thus, two motivations–to identify kinases involved in transmitting stress signals and to explore roles for kinases that are contingent on the environment–prompted us to generate a dataset in which we measured global gene expression in a panel of kinase mutant yeast strains across a battery of environmental conditions.

## Results

### Measurement of global gene expression in 28 kinase mutants in ten environmental conditions

With the goal of understanding how environmental stress signals propagate through kinase pathways to alter gene expression, we constructed a set of 28 yeast strains harboring mutations in kinases implicated in stress response signaling (Fig 1A and 1B). In each strain, an endogenous kinase gene was replaced with an analog sensitive (AS) allele. The AS alleles encode a key "gatekeeper" mutation designed to preserve catalytic function while enabling the kinase to be inhibited by addition of a cell-permeable ATP analog [16]. For eight of the kinases, the gatekeeper mutant had not been previously generated or validated (Ksp1, Mrk1, Rim11, Rim15, Ssn3, Ste11, Yak1 and Ygk3). We did not develop assays to validate these conditional mutations in this study, so it is possible that the bioinformatically-defined gatekeeper mutations may not confer analog sensitivity to these kinases.

We grew the panel of mutant kinase strains along with four wild type replicates to exponential phase in rich media, added a cocktail of three ATP analogs to ensure efficient inhibition of the various kinases, and subjected this set of 32 strains to 10 different environments: rich media (YPD), synthetic media (SDC), heat shock (39°C), hyperosmotic shock (0.5 M NaCl), glucose depletion (YP osmo-balanced with sorbitol), endoplasmic reticulum (ER) stress (5 μg/ml tunicamycin), oxidative stress (250 μM menadione), proteotoxic stress (10 mM azetidine 2-carboxylic acid (AZC)), target of rapamycin (TOR) inhibition (1 μg/ml rapamycin), and antifungal drug exposure (250 μg/ml fluconazole). We harvested cells following 20 minutes in each environment, purified total RNA, and performed polyA+ RNA sequencing, generating a total of 301 deeply sequenced transcriptome-wide datasets (> 25X genome coverage for all samples). Since stress responses are inherently transient, we chose the 20-minute timepoint to allow for enough time for the stress to be perceived and lead to transcriptional changes, but before the responses were attenuated. Due to sequencing constraints, we did not perform experiments in the absence of inhibitor cocktail nor were we able to perform biological replicates for the mutant strains in each condition, limiting our statistical power for any one gene in any mutant in a given condition. However, the four replicates of wild type in each condition provided us with high statistical power for the expression of each gene in the genome in all conditions in the reference strain. While we sequenced RNA from each mutant strain 10 times

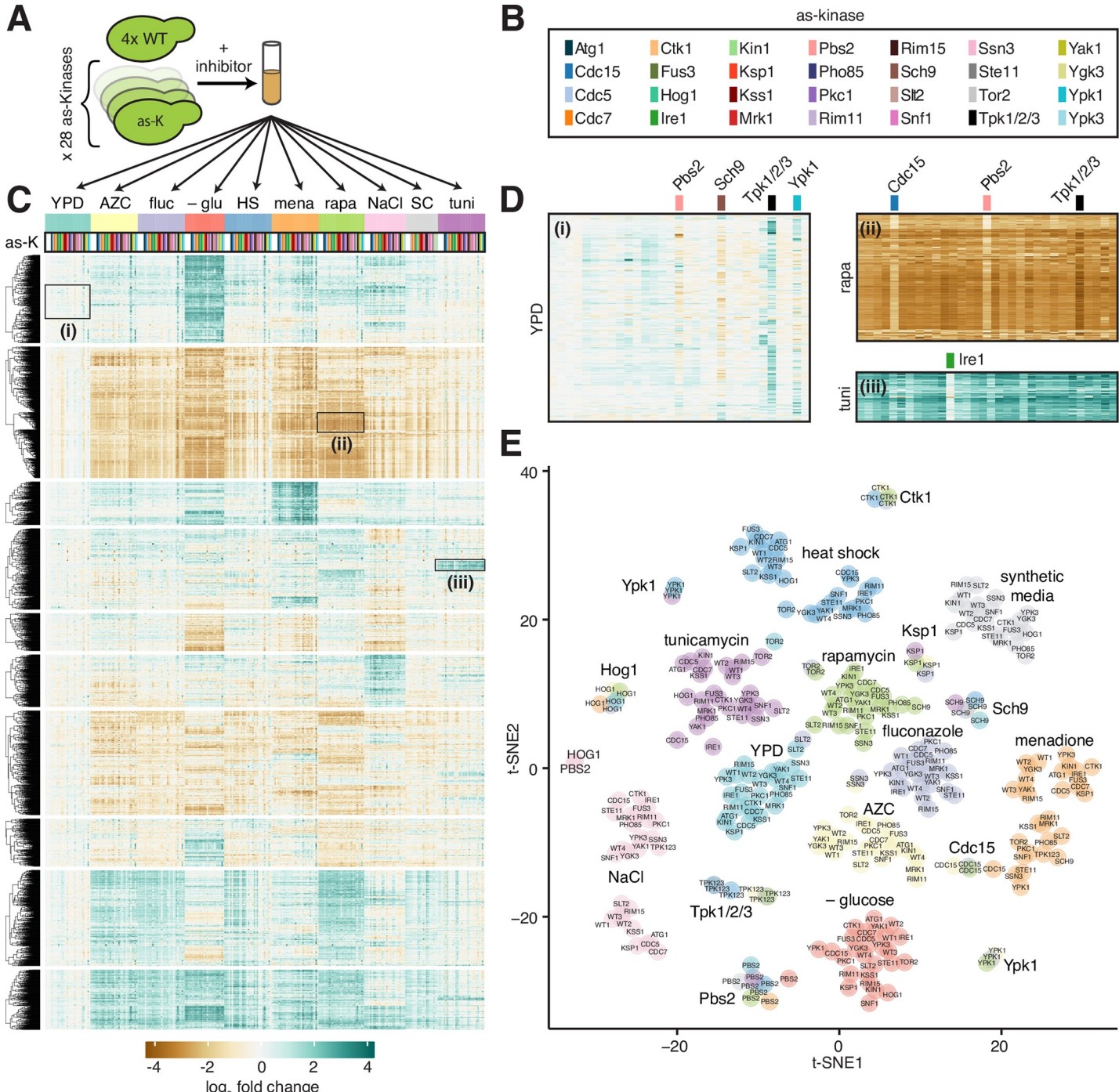

**Fig 1. Measurement of global gene expression in 28 kinase mutants in ten environmental conditions. A)** Schematic of experimental protocol. Four replicates of a wild type yeast strain and 28 isogenic strains harboring point mutations in genes encoding kinases that render the kinases "analog sensitive" (AS, see methods) were grown to exponential phase, treated with inhibitor cocktail for 5 min, and subjugated to one of 10 environmental conditions for 20 minutes. **B)** Color key indicating AS kinase strains in (c), (d) and (e). **C)** Expression heatmap of all genes across all samples in the dataset relative to the level of each gene in wild type cells in YPD. Gene rows are clustered hierarchically, samples are ordered by environmental condition and by AS kinase within each condition alphabetically. **D)** Expanded expression heatmaps for three regions: i) inhibition of Pbs2, Sch9, Tpk1/2/3, and Ypk1 resulted in altered expression of genes enriched for ribosome biogenesis factors in YPD; ii) inhibition of Cdc15, Pbs2 and Tpk1/2/3 altered levels of a set of genes induced by rapamycin enriched for alternative metabolic enzymes; iii) inhibition of Ire1 attenuated induction of genes enriched for ERAD components and UPR targets in tunicamycin. **E)** Clustering of RNA-seq samples following dimensionality reduction by PCA followed by t-SNE plotted on a two-dimensional projection. Environments are color-coded.

across the conditions, our choice to sacrifice biological replicates for the mutants in each environment enabled us to broadly survey conditions.

We aligned reads, quantified read counts per gene [17] and used DESeq2 to generate normalized expression values (see methods) [18]. With this processed data, we calculated the $\log_2$ fold change of each gene in each sample with respect to wild type cells in YPD plus inhibitor cocktail. Hierarchical clustering revealed a structured matrix ordered with respect to environmental perturbations in the horizontal direction (Fig 1C). The genes formed ten clusters in the vertical direction, with the top two clusters enriched for stress response genes and ribosome biogenesis genes, respectively (see S1 Table for gene ontology terms associated with each cluster). Inhibition of certain kinases also induced structured patterns in certain environments. For example, zooming into the top cluster enriched for stress response genes, it is evident that inhibition of Pbs2-as, Sch9-as, Tpk1/2/3-as, and Ypk1-as altered expression of many genes in the absence of any environmental perturbation (Fig 1D-i). In the cluster enriched for ribosome biogenesis genes, inhibition of Cdc15-as, Pbs2-as, and Tpk1/2/3-as followed by treatment with rapamycin resulted in dysregulation of many genes (Fig 1D-ii). Another example is Ire1-as, inhibition of which had an effect on genes involved in ER stress specifically in tunicamycin (Fig 1D-iii).

To visualize similarity among the samples across the full dataset, we performed principal component analysis and t-distributed stochastic neighbor embedding (t-SNE) (Fig 1E) [19]. In this low dimensional representation of the data, samples that are correlated across the transcriptome should group together. In general, we observe that clusters form among samples exposed to the same environmental perturbations. However, additional clusters formed among several groups of samples in which a common kinase was inhibited–irrespective of environment–such as Tpk1/2/3-as (homologs of protein kinase A) and the TOR pathway kinases Sch9-as and Ypk1-as. The t-SNE analysis serves to underscore the above observation that environmental perturbations have a dominant effect on gene expression. Notably, however, inhibition of Tpk1/2/3 and TOR pathway kinases resulted in transcriptome-wide effects with comparable magnitude to environmental perturbations.

## Identification of kinase mutants that alter environment-specific gene expression

We next interrogated the dataset to determine which kinases contribute to gene expression in each environmental condition. Specifically, we asked which AS kinase strains displayed gene expression patterns that differed from wild type in each environment. To this end, we used the wild type replicates to define sets of differentially expressed genes in each environment compared to YPD (S2 Table, see methods). For these sets of differentially expressed genes, we determined the fold change of each gene in the AS kinase strains relative to the average wild type value under the same environmental perturbation.

First, we plotted these two quantities against each other for every AS mutant in ER stress (tunicamycin) (Fig 2A). We then used linear regression to apply a line of best fit to indicate trends in the levels of the differentially expressed genes. For the set of >1000 genes differentially expressed in tunicamycin, many AS kinases such as Cdc7-as, Ctk1-as and Fus3-as produce a response that is indistinguishable from that of wild type (flat regression line, few outliers). However, mutants such as Cdc15-as and Ire1-as show an attenuated response compared to that of the wild-type (regression lines with negative slopes), while Cdc5-as, Kin1-as and Ksp1-as show enhanced differential expression (regression lines with positive slopes). The attenuated tunicamycin response in the Ire1-as cells is consistent with a known role for Ire1 in the unfolded protein response [6] but the other implicated kinases have not been previously

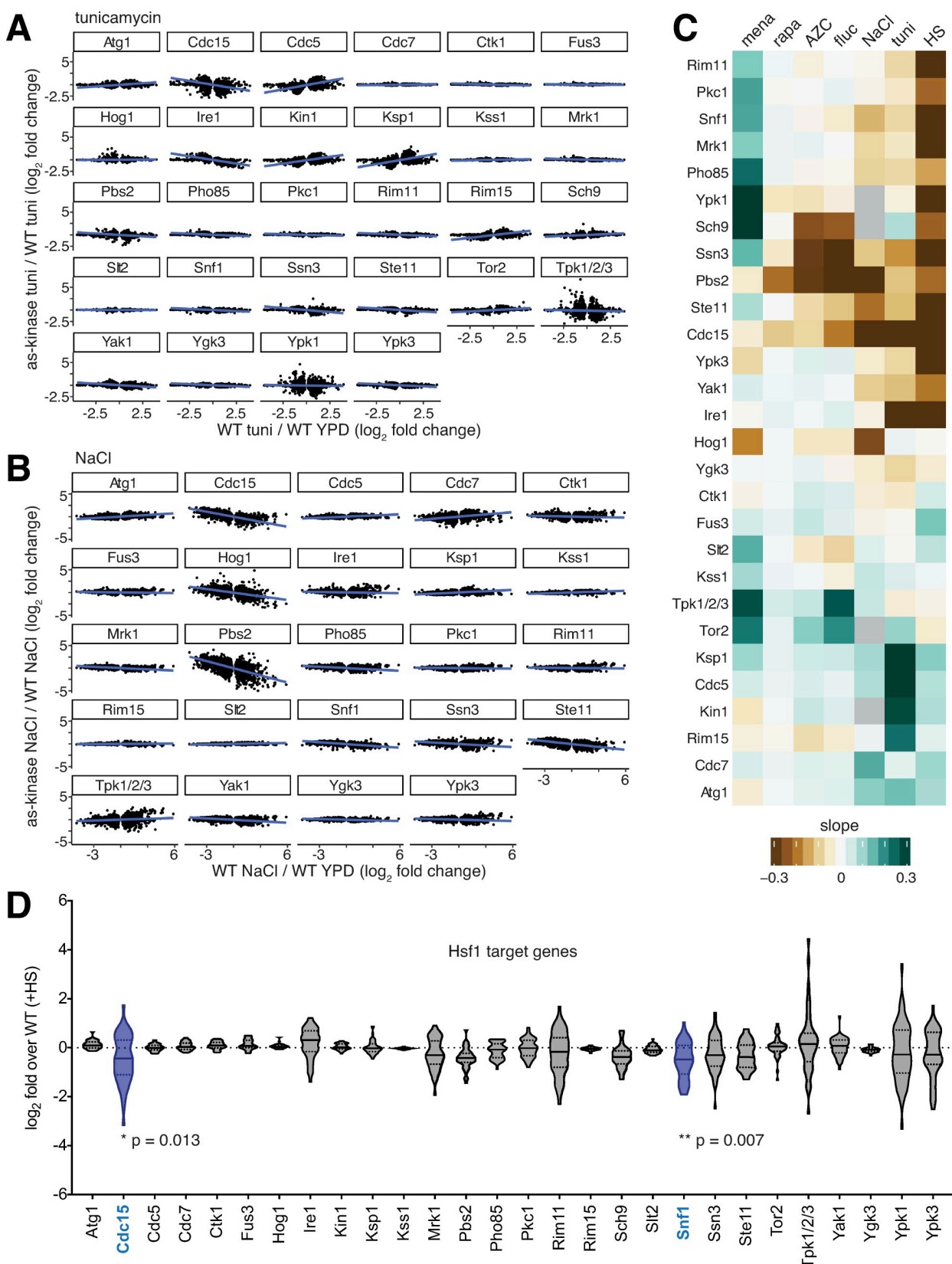

Fig 2. **Identification of kinase mutants that alter environment-specific gene expression. A)** Genes differentially expressed in tunicamycin relative to YPD in wild type cells plotted for each inhibited kinase. The x-axis is the $\log_2$ fold change for the wild type in tunicamycin; the y-axis is the $\log_2$ fold change for the inhibited kinase strain relative to wild type. **B)** the same as (A), but for the NaCl. **C)** Slopes of regression lines fitted to plots like those in (A) and (B) for all inhibited kinases in all environments (missing samples in gray). **D)** Expression level following heat shock of the 42 genes controlled by Hsf1. Violin plots show the $\log_2$ fold change of the 42 genes in heat shock relative to wild type in heat shock for the set of inhibited kinases. Distributions of expression levels were analyzed pairwise by two-way ANOVA, revealing that Cdc15-as and Snf1-as show statistically significant attenuation of the Hsf1 operon in heat shock.

associated with ER stress. An additional group of AS kinases including Tpk1/2/3-as and Ypk1-as have flat slopes but large numbers of outliers, indicating low correlation with wild type across the set of differentially expressed genes. This apparent global dysregulation suggests that inhibition of these kinases may impair the ability of the cell to sense or respond to tunicamycin.

Similar analysis for hyperosmotic stress (NaCl) revealed that Pbs2-as, Hog1-as, Ste11-as and Cdc15-as have negative slopes, while Cdc7-as and Tpk1/2/3-as have positive slopes (Fig 2B). Ste11, Pbs2 and Hog1 form a linear mitogen activated protein kinase (MAPK) cascade that is activated by hyperosmotic stress, while Tpk1/2/3 is known to be inactivated under a variety of stress conditions including hyperosmotic shock [2]. The mitotic cell cycle regulators Cdc15 and Cdc7 have not been previously studied in the context of osmo-signaling.

We plotted the slopes for relationships as above for all inhibited kinases in each of the environmental perturbations as a heat map (Fig 2C). The set of kinases included in this study was enriched for those likely to be involved in the heat shock response. Indeed, while several AS mutants altered expression in many of the environments, 14/28 AS mutants attenuated the response to heat shock. However, only two mutants, Cdc15-as and Snf1-as, displayed significantly altered expression of the set of 42 genes controlled by the heat shock transcription factor Hsf1 (Fig 2D) [20]. Cdc15 has no known link to Hsf1, but Snf1 has been previously reported to phosphorylate and activate Hsf1 [21]. This relative dearth of interactions between Hsf1 targets and kinases is consistent with the observation that a mutant of Hsf1 lacking all phosphorylation sites can still be induced by heat shock [22]. Thus, rather than influencing Hsf1 activity, the other 12 kinases that modulate the heat shock transcriptome do so by affecting the levels of the >400 mRNAs that are differentially expressed during heat shock in an Hsf1-independent manner [20].

## Kinase mutants that alter common environmental responses do so by affecting overlapping and distinct sets of genes

The above analysis revealed that multiple kinase mutants alter differential gene expression in most environments (Fig 2C). However, it is unclear if these kinases are acting on the same sets of target genes. To ascertain this for the case of tunicamycin, we performed hierarchical clustering of the differentially expressed genes defined in Fig 2A and identified seven clusters (Fig 3A). Gene ontology analysis revealed enriched categories in five clusters. We examined Cdc15-as and Ire1-as since they both attenuated the response to tunicamycin (Fig 2A). While both mutants showed reduced expression of the "regulatory processes" and increased expression of the "ribosome biogenesis" clusters, only Ire1-as showed reduced expression of the "response to ER stress" cluster. In contrast to Ire1-as, inhibition of Cdc15-as did not exclusively alter any set of genes. However, while not unique to Cdc15-as, inhibition of Cdc15 showed strong repression of the amino acid metabolism genes in both tunicamycin and heat shock (Fig 3A). With respect to the genes differentially expressed in tunicamycin, when compared to each other across the set of environmental conditions, Cdc15-as and Ire1-as closely mirrored each other except for the "response to ER stress" cluster in response to tunicamycin

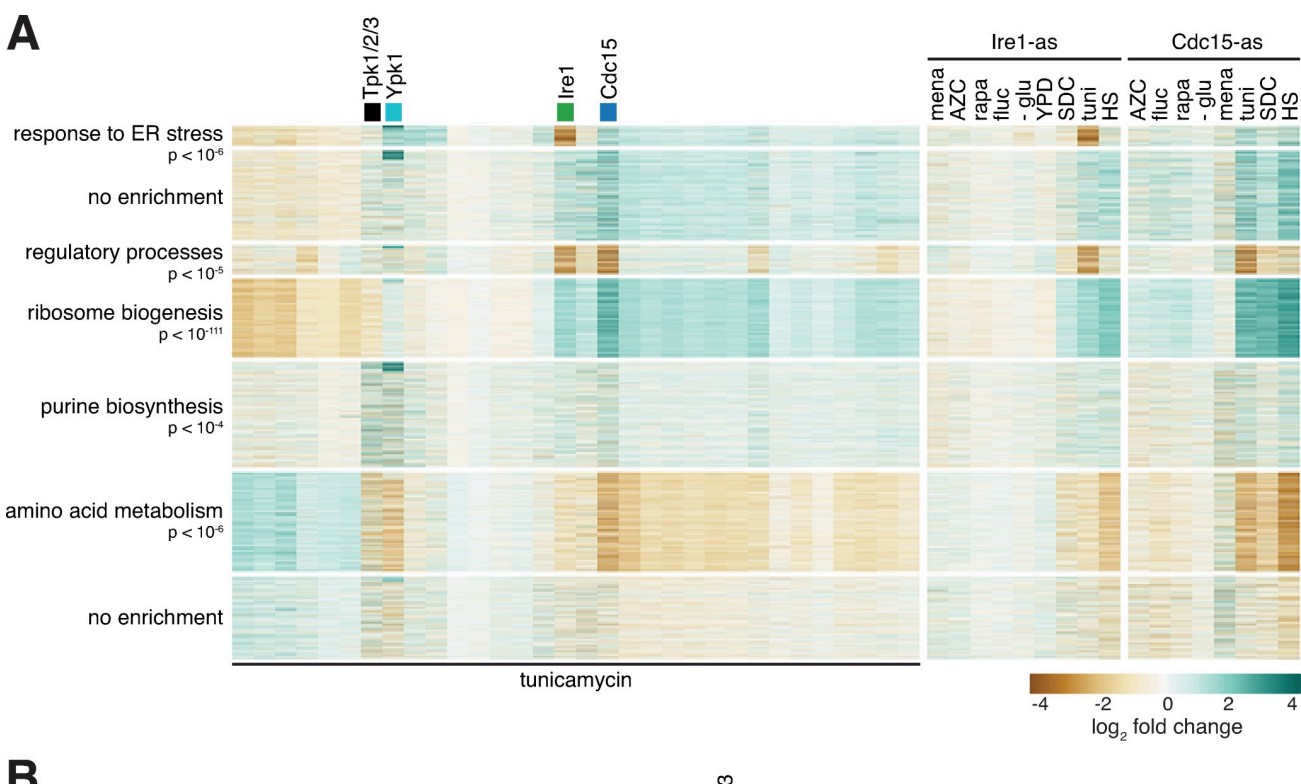

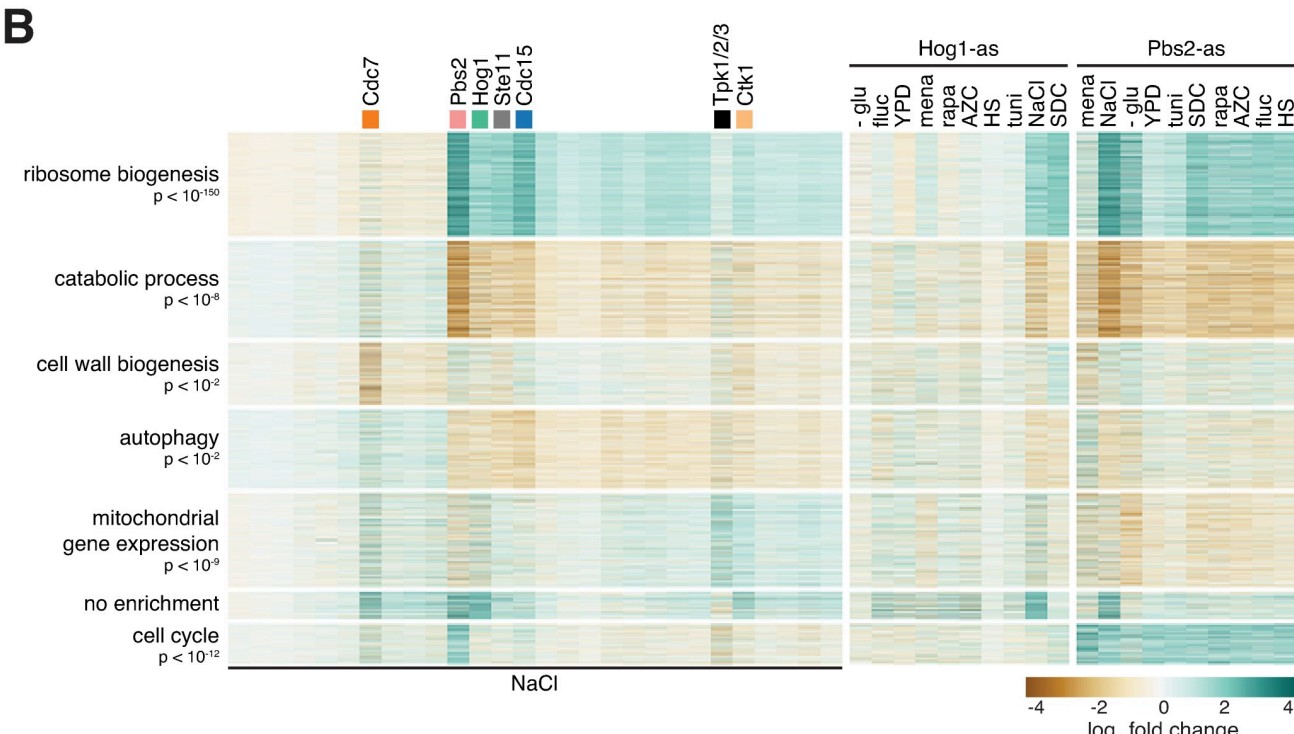

**Fig 3. Kinase mutants that alter common environmental responses do so by affecting overlapping and distinct sets of genes. A)** Heatmaps showing the set of genes differentially expressed in tunicamycin in wild type cells, clustered into seven groups labeled by the most significant gene ontology term enriched in the group. Left panel: $\log_2$ fold change relative to wild type in tunicamycin for the set of kinase mutants treated with tunicamycin. Middle and right panels: For Ire1-as and Cdc15-as, $\log_2$ fold change of the same set of genes across all environments, shown relative to wild type in each environment. **B)** As in (A) but for set of genes differentially expressed in in NaCl.

(Fig 3A, right). This suggests that Cdc15 and Ire1 may impinge on a common generic pathway, but that only Ire1 plays a specific role in response to tunicamycin.

In the hyperosmotic stress environment, we found that the Pbs2-as, Hog1-as, Ste11-as and Cdc15-as, display a common pattern of altered expression across the genes differentially expressed in NaCl (Fig 3B). With the exception of Cdc15-as, these kinases are known to form a linear MAPK cascade [9, 23–25]. In addition, Cdc7-as uniquely shows altered expression of cell wall biogenesis genes (Fig 3B). When compared to each other across all the environments using the genes differentially expressed in NaCl, Hog1-as and Pbs2-as were similar but not identical, with the magnitude of the effect stronger for Pbs2-as (Fig 3B, right). This could be due to Hog1-independent roles for Pbs2 or to differential penetrance of the AS alleles.

## Identification of nonlinear interactions between environmental perturbations and inhibited kinases

To systematically identify interactions between particular environments and inhibition of specific kinases that affect the expression of individual genes, we constructed a mathematical model for the expression of each gene in the genome. In the model, the expression level of a gene is determined by the sum of two effects assumed to be independent: the environmental perturbation and the kinase inhibition ($\Delta e = c_i + k_j$, where **e** is the expression of a given gene, **c** is the contribution of the environmental condition (indexed i) to the expression of the gene, and **k** is contribution of the inhibited kinase (indexed j) to the expression of a gene. When a predicted $\Delta e$ from this model–i.e., a value for the $\log_2$ fold change for a given gene in a specific environmental condition with a particular kinase inhibited–is significantly different from the measured $\log_2$ fold change, this indicates a nonlinear interaction between the perturbation and the inhibition. This nonlinearity implies that the activity of the kinase may be somehow involved in the environmental response in a manner analogous to genetic interaction epistasis analysis [26, 27].

To apply the model to the data, we decomposed the full gene expression dataset using ordinary least squares regression (Fig 4A, left, see methods). To compute the contribution of an environmental condition ($c_i$) to expression, we determined the fold change for each gene for the wild type strain in this environment relative to expression in YPD for the wild type strain (Fig 4A, middle). To compute the contribution of a kinase ($k_j$), we determined the fold change in every gene in the corresponding AS kinase strain in YPD relative to wild type in YPD (Fig 4A, right). For every gene, we correlated the sum of these two terms to the experimentally measured fold change value for each of the >300 measurements made for that gene across the dataset, estimating the $R^2$ value of this correlation. This analysis generated $R^2$ values for >6000 genes, whose distribution can be examined to assess the extent to which such a linear model broadly applies. (Fig 4B). The mean of this distribution is 0.74, indicating that, on average, the linear model can explain 74% of the variance for a given gene (see Residual and Model performance tab on online applet for all gene models: https://mace.shinyapps.io/kinase-app/).

Gene expression measurements that the model predicts well require no further explanation. However, large error between a model prediction and an experimental measurement suggests that the environmental perturbation is not independent of the inhibited kinase. We compared the distribution of the model errors to a normal distribution using a quantile-quantile plot, which demonstrated that the error was approximated well by normal distribution until the error in the model exceeded 2 standard deviations of the mean (Fig 4C). This error–the difference between predicted and measured expression values–is termed the "residual" and will be quantified in units of standard deviation from the mean ($\sigma$). We calculated the residual for all >2 x $10^6$ data points and plotted the distribution as a density contour plot to demonstrate that

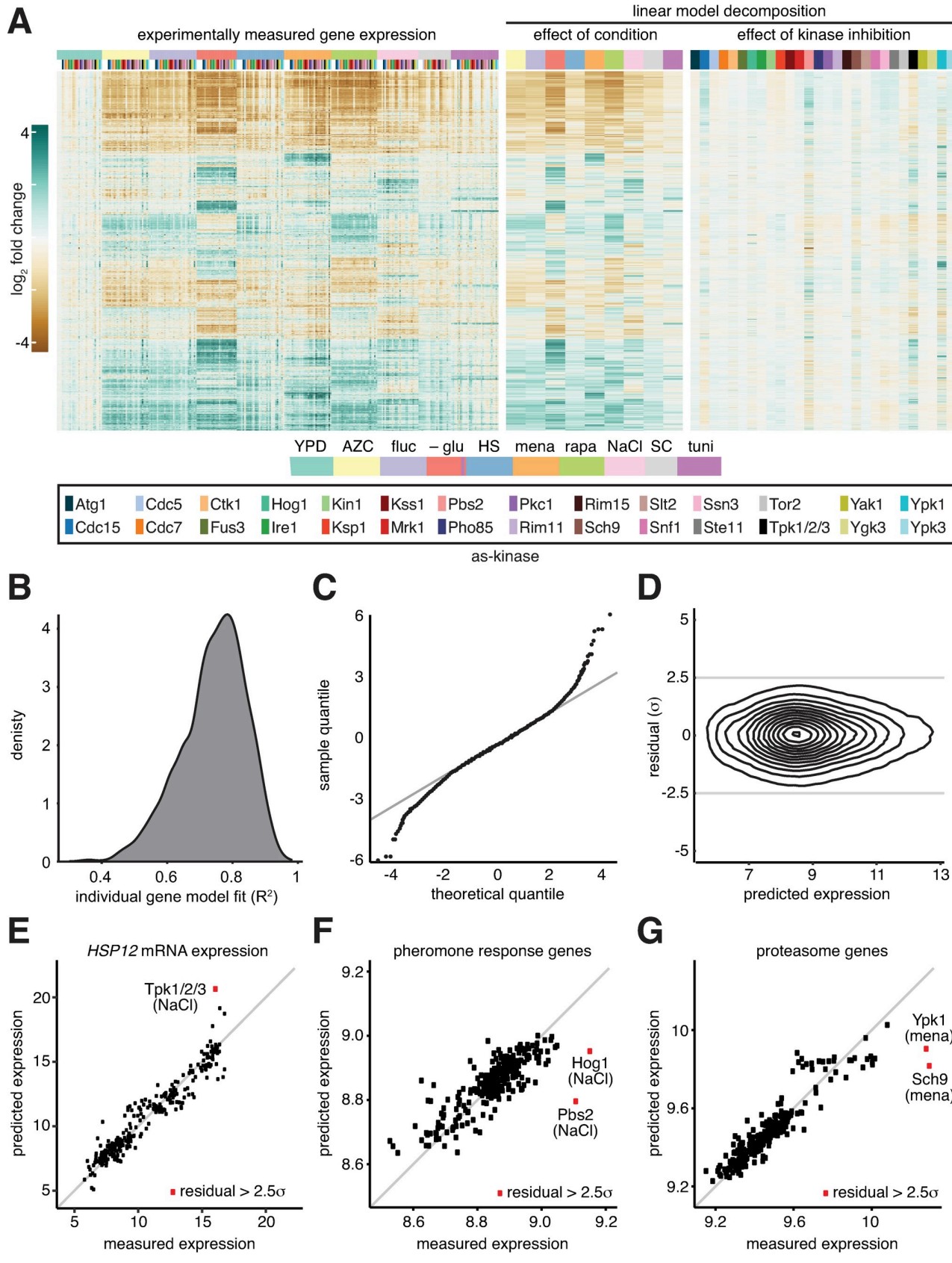

**Fig 4. Identification of nonlinear interactions between environmental perturbations and inhibited kinases. A)** Linear decomposition of the gene expression matrix. Left panel: Measured gene expression matrix ($\log_2$ fold change of each gene in each environment relative to the expression in wild type in YPD; same data as Fig 1C). Middle: A linear model was fit to each gene to estimate the contribution of each environment on expression in the wild type. The heat map shows the estimated $\log_2$ fold change for each gene relative to expression in wild type in YPD. Right: A linear model was fit to each gene to estimate the contribution of inhibiting each kinase in YPD. The heat map shows the estimated $\log_2$ fold change for each gene relative to expression in wild type in YPD. Kinases and environments are color-coded as depicted in the key at the bottom. **B)** Frequency distribution of the set of individual gene models, binned by the quality of the fit across the full set of measurements–i.e., how well each model matches the measured expression of the given gene across the set of >300 samples–quantified as the $R^2$. **C)** Quantile-quantile plot comparing the distribution of the residuals (the differences between the model predictions and measured values) against a theoretical normal distribution. 98% of the data points lie on the x = y line, suggesting that these residuals are normally distributed and therefore unlikely to be meaningful (just modeling and measurement error). The residuals of the remaining 2% of samples are no longer normally distributed, suggesting the linear model is insufficient in these cases. **D)** Density contour plot of all two million model residuals as a function of absolute gene expression, demonstrating low correlation between residuals and expression level. **E)** Scatter plot of all measurements of the expression of *HSP12*. For each sample, measured *HSP12* expression is plotted versus the predicted expression value. For a single sample, Tpk1/2/3-as under NaCl, the absolute value of the difference between the predicted and measured expression values exceeded 2.5 standard deviations. Note: *HSP12* was not measured in Tpk1/2/3-as following glucose depletion. **F)** Scatter plot of all measurements of the mating pheromone response genes versus model predictions. For each sample, the expression of all genes with the gene ontology term "pheromone response" was averaged. For Hog1-as and Pbs2-as under NaCl, the measured expression of the pheromone response genes exceeded the model predictions by greater than 2.5 standard deviations. **G)** As in (F), but for genes with the gene ontology term "proteasome". For Ypk1-as and Sch9-as under menadione, the residuals exceeded 2.5 standard deviations.

the vast majority of the data points (>98%) are within 2.5σ and that the residuals are not correlated to average gene expression (Fig 4D). We set a threshold for residuals > 2.5σ to identify candidate transcripts regulated by a particular kinase in a given environment (S3 Table).

## Instances of residuals with putative biological significance

One of the genes that was among those best fit by the model across the full dataset was *HSP12* ($R^2 > 0.9$), a stress-induced gene predominately activated by the paralogous general stress transcription factors Msn2 and Msn4 (Msn2/4) [1, 2, 28, 29]. However, one data point was an outlier that was poorly explained by the model and had a residual > 2.5σ: Tpk1/2/3-as cells treated with NaCl (Fig 4E). In this case, the model predicted higher *HSP12* expression than what we measured. This indicates that Tpk1/2/3-as inhibition and NaCl do not exert independent effects on *HSP12* expression and suggests that NaCl may induce *HSP12* in wild type cells via inactivation of Tpk1/2/3, as has been previously shown experimentally [30]. We would expect the same result for glucose depletion in Tpk1/2/3-as cells, but this sample was not in our dataset.

We next extracted all genes with residual of > 2.5σ and performed gene ontology analysis (go slim terms) (S4 Table). One of the top categories produced by this analysis was "response to pheromone". We averaged expression for all genes with this designation and plotted the predicted versus measured expression for all data points (Fig 4F). Two points were poorly predicted by the model: Hog1-as and Pbs2-as in NaCl. In both cases, the measured expression of the pheromone response regulon was greater than predicted by the model. This result recapitulates a classical finding of crosstalk between MAPK signaling pathways. The pheromone and high osmolarity glycerol (HOG) pathways share a common upstream regulator (Ste11) [9, 24, 25, 31]; in the presence of hyperosmotic shock, Hog1 is required to prevent spurious activation of the mating program [25].

A second enriched gene set contained genes encoding proteasome subunits in cells exposed to menadione (oxidative stress). As we did for the pheromone response genes, we averaged expression of all genes with a "proteasome" GO term and plotted the predicted expression in each of the samples as a function of experimentally measured expression (Fig 4G). This plot reveals that the proteasome is highly induced by menadione across all samples, consistent with prior studies [32]. Moreover, proteasome gene expression was very poorly predicted in response to menadione for two AS kinases: Sch9-as and Ypk1-as. When these kinases are

inhibited, the proteasome genes are even more highly induced by menadione than predicted. This suggests that Sch9 and Ypk1 may serve to dampen proteasome gene induction in response to menadione in wild type cells.

## Prediction of links between kinases and transcription factors

To implicate transcription factors in responding to specific kinases in particular environments, we determined all genes with significant residuals in each sample and used MEME [33] to search for transcription factor (TF) binding motifs that are enriched upstream of these genes (S5 Table). Across this gene set–which contains the outliers from the linear model–we identified enriched binding motifs associated with 33 different TFs using the JASPAR database [34]. For 21/28 mutant kinases in our dataset, this analysis identified a putative connection to at least one TF in at least one condition. Kinases such as Ypk1, Ypk3 and Rim11 connect to multiple TFs under heat shock conditions, while Snf1 connects only to a single TF during heat shock (Fig 5A). Similarly, Sch9 and Ssn3 connect to many TFs under oxidative stress, while Ctk1 and Ire1 have a single TF link (Fig 5B). It remains to be determined how many of these links represent direct kinase-substrate interactions.

While most of the conditions had sparse connections between kinases and TFs, heat shock and menadione had complex interaction networks consistent with the pleiotropic nature of temperature and oxidative stresses (Fig 5A and 5B). Heat shock had six kinases interacting with ten TFs, while menadione had nine kinases interacting with 15 TFs. Notably, while both heat shock and menadione led to enrichment of TFs known to be involved in environmental stress responses (Msn2/4, Dot6/Tod6, Stb3, Sfp1) [2, 28, 35], the kinases that connect to these TFs are largely non-overlapping (with the exception of Ypk1-as and Cdc15-as). This analysis also recapitulated known interactions, such as that between Ire1 and the UPR TF Hac1 in tunicamycin [6, 8], and predicted novel regulatory connections, such as that between Cdc15 and Msn2 in response to tunicamycin. Taken together, these results demonstrate that residual analysis can be used to suggest previously unknown pathways that connect environmental conditions to gene expression through kinases and their responsive TFs.

## The HOG pathway is an upstream negative regulator of Tpk1/2/3 under all conditions

In contrast to the instances of environment-specific interactions between kinase inhibition and gene expression described above, we observed that across all conditions, Tpk1/2/3-as and Pbs2-as affect the same genes across the dataset, but do so in opposite directions for many genes (Fig 6A). Tpk1/2/3 is the master kinase of the environmental stress response (ESR) [2]. The ESR is comprised of two branches: the pan-stress induced (iESR) and pan-stress repressed (rESR) gene modules [2]. Mechanistically, Tpk1/2/3 has been shown to directly control phosphorylation of two sets of paralogous TFs–Msn2/4 and Dot6/Tod6 –that induce the iESR and repress the rESR, respectively [2, 36–38]. Phosphorylation by Tpk1/2/3 inactivates all four of these TFs by preventing their nuclear localization [28, 35]. To compare the effects of inhibition of Tpk1/2/3 to inhibition of Pbs2 with respect to the ESR, we extracted the $\log_2$ fold change of the iESR and rESR genes for all experiments where Tpk1/2/3-as was inhibited; we did the same for the values of the iESR and rESR genes when Pbs2-as was inhibited. For reference, we extracted the $\log_2$ fold change for the iESR and rESR genes when all other kinases in dataset were inhibited. An overlay of the distributions of the Tpk1/2/3-as, Pbs2-as, and reference sets shows that: 1) inhibition of Tpk1/2/3-as led to induction of the iESR and repression of the rESR; 2) inhibition of Pbs2-as had the opposite effect: repression of the iESR and induction of the rESR; 3) inhibition of other kinases minimally affected the iESR and rESR genes (Fig 6B

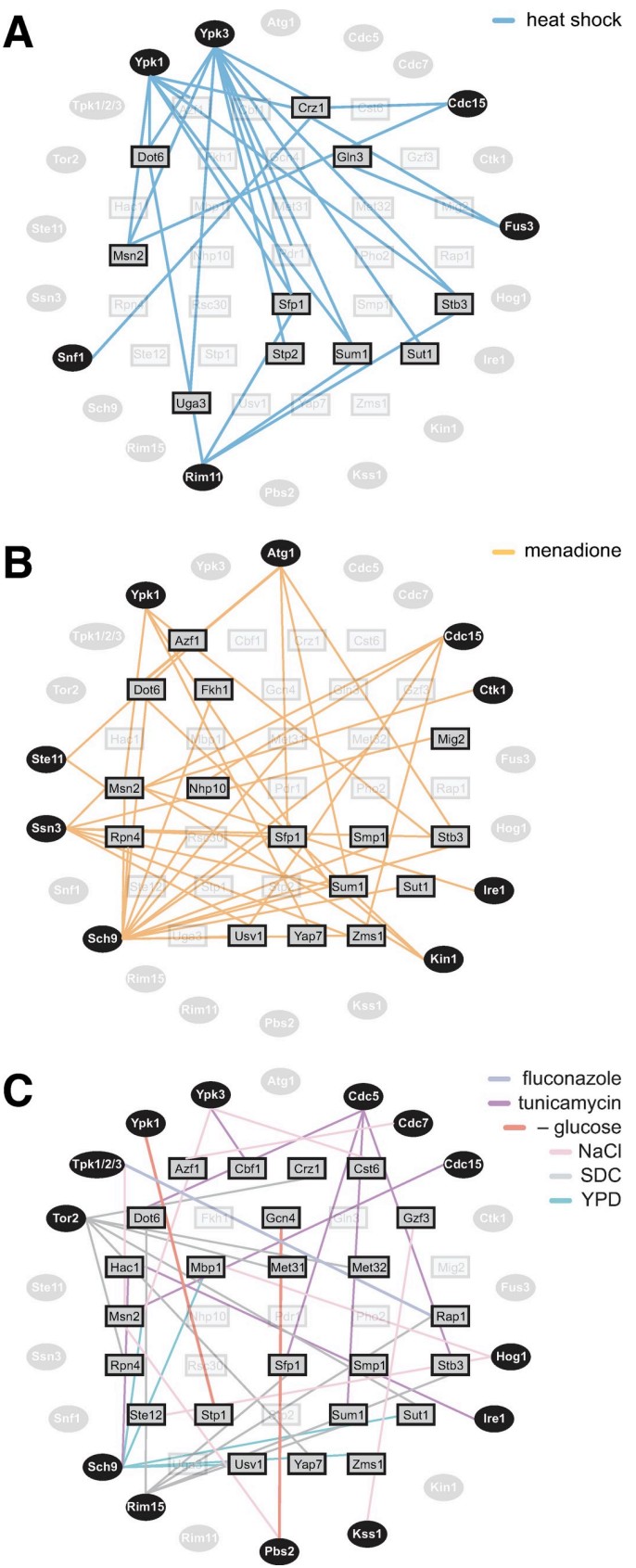

**Fig 5. Prediction of links between kinases and transcription factors. A)** The set of all genes with expression measurements with model residuals greater than an absolute value of 2.5 standard in any sample was analyzed for enriched TF binding sites in their promoter regions. The edges on the graph connect peripheral AS kinases with interior TFs. Connections indicate that when the given kinase is inhibited during heat shock, the genes with significant residuals are enriched for binding sites for the given TF. **B)** As in (A), but for menadione. **C)** As in (A), but for all environments except heat shock and menadione.

and 6C). This suggests that Tpk1/2/3 and Pbs2 regulate the same pathway but in an antagonistic way.

To experimentally test this prediction and to order the pathway, we utilized flow cytometry to measure a canonical fluorescent reporter of the iESR, *HSP12pr*-GFP [39–41]. We first validated that addition of 0.5 M NaCl and inhibition of Tpk1/2/3-as with 5 μM 1NM-PP1 induced the *HSP12pr*-GFP reporter (Fig 6D). If Pbs2 is involved in regulation of *HSP12pr*-GFP, then its deletion should alter *HSP12pr*-GFP levels. Indeed, we observed that in *pbs2Δ* cells, the basal level of *HSP12pr*-GFP was reduced 4-fold. Moreover, *pbs2Δ* obviated the ability of NaCl to induce the reporter, indicating that Pbs2 plays positive role in both setting the basal level of *HSP12pr*-GFP and inducing it in response to NaCl (Fig 6D). If Pbs2 is acting downstream of Tpk1/2/3, then direct inhibition of Tpk1/2/3-as with 1NM-PP1 should not be able to rescue induction of *HSP12pr*-GFP in *pbs2Δ* cells. However, we observed that addition of 1NM-PP1 led to full induction of the reporter, suggesting that Pbs2 must be acting upstream of Tpk1/2/3. To verify this, we expressed a constitutively active version of Ras2 (Ras2-G19V), a known upstream activator of Tpk1/2/3. Activation of Pbs2 by NaCl failed to induce *HSP12pr*-GFP in the presence of Ras2-G19V, while direct inhibition of Tpk1/2/3-as with 1NM-PP1 rescued induction of the reporter in the presence of Ras2-G19V. Thus, Pbs2 must be acting to inhibit Tpk1/2/3 at or above the level of Ras activation.

Tpk1/2/3 regulation is implemented by tuning the ratio of free kinase to repressor-bound kinase via the second messenger cAMP [42]. Since Pbs2 is acting at or above the level of Ras2, we hypothesized that in the absence of Pbs2, the ratio of free Tpk1/2/3 to repressed Tpk1/2/3 would be higher, and it would therefore take a higher concentration of inhibitor to inactivate Tpk1/2/3-as in *pbs2Δ* cells compared to otherwise wild-type cells. To test this, we performed a 1NM-PP1 dose response assay and measured the *HSP12pr*-GFP reporter in Tpk1/2/3-as/*pbs2Δ* cells (Fig 6E). Consistent with our hypothesis, Tpk1/2/3-as/*pbs2Δ* cells were desensitized to 1NM-PP1 relative to Tpk1/2/3-as cells (otherwise wild type). Moreover, Tpk1/2/3-as/*hog1Δ* cells phenocopied the Tpk1/2/3-as/*pbs2Δ* cells and were equally desensitized to 1NM-PP1. Taken together, these data indicate that the HOG pathway is a general upstream negative regulator of Tpk1/2/3 (Fig 6F).

## Discussion

The kinase domain evolved a highly plastic structure that enables it to maintain its core catalytic function (transferring a phosphate group from ATP onto a substrate protein) while radiating into a family that includes > 500 members encoded in human cells, each receiving a specific set of inputs and relaying information to a distinct set of substrates [43]. Like wires in electronic devices, kinases transmit information using a common currency–phosphorylation–that enables them form serial and parallel relays. But unlike static wires, kinases form dynamic networks that rearrange their spatial localization and interaction partners in complex patterns that are contingent on the environment. To gain insight into the plasticity of a subset of the cellular "kinome", we generated transcriptomic data from yeast cells exposed to 10 different environments in the context of mutations in 28 different kinases (21% of total kinases in

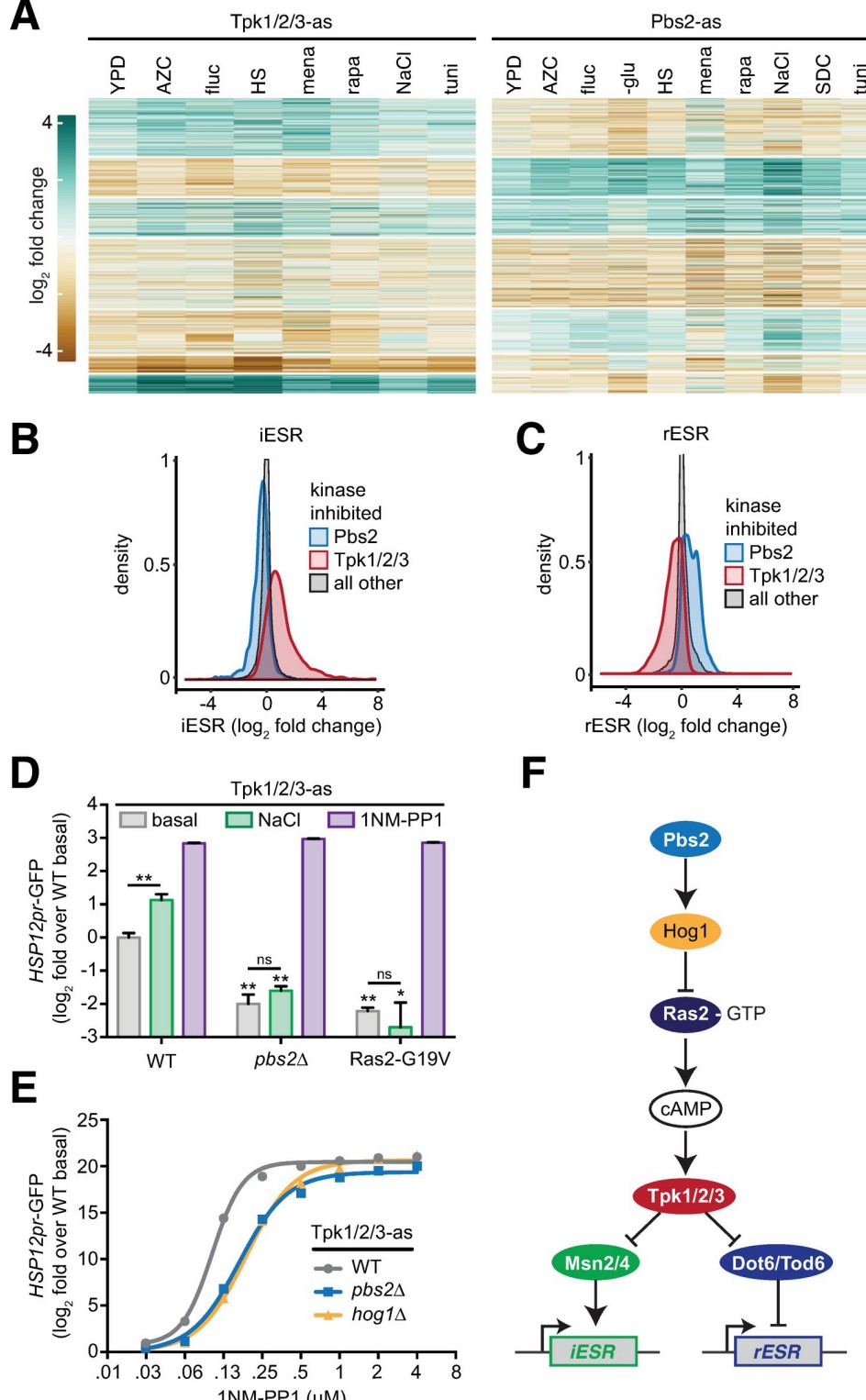

**Fig 6. The HOG pathway is an upstream negative regulator of Tpk1/2/3 under all conditions. A)** Heatmaps showing the genome-wide effects of inhibition of Tpk1/2/3-as and Pbs2-as relative to wild type across the set of environments. **B)** Distribution of fold changes for all genes in the iESR in Pbs2-as cells, Tpk1/2/3-as cells and all other strains. **C)** As in (B), but for the rESR. **D)** A GFP reporter driven by the *HSP12* promoter was measured by flow cytometry under basal conditions, following 0.8 M NaCl for 1 hour and following addition of 5 μM 1NM-PP1 for 1

hour in Tpk1/2/3-as cells, Tpk1/2/3-as/*pbs2Δ* cells and Tpk1/2/3-as/Ras2-G19V cells. The median of $10^4$ cells was measured in triplicate for each condition, and the error bar represents the standard deviation of the replicates. **E)** Tpk1/2/3-as, Tpk1/2/3-as/*pbs2Δ* and Tpk1/2/3-as/*hog1Δ* cells were subjected to a 1NM-PP1 dose response assay, using the *HSP12p*r-GFP as a readout. Each concentration of inhibitor was incubated with cells for 1 hour. Measurements performed analyzed as in (D). **F)** Proposed architecture of the upstream signaling pathway controlling the ESR.

yeast). We chose this set of kinases to be enriched for those with known or putative functions in stress pathways.

Transcriptional responses to a large number of environmental perturbations have been measured in yeast cells. A major conclusion from these studies is that across nearly all environmental stress conditions, cells display a common gene expression signature known as the environmental stress response (ESR) [2, 37, 38, 44]. The ESR is comprised of two sets of genes that cells coordinately regulate across stresses: one that gets repressed (enriched for ribosomal protein and biogenesis genes), and one that gets induced (enriched for genes involved in alternative carbon and nitrogen metabolism) [2, 44]. Subsequent studies have traced the regulation of the ESR to two major signaling axes: Ras/PKA and TOR. Key regulatory kinases in these pathways (Tpk1/2/3 and Tor1/2, respectively) phosphorylate and directly regulate the activity of transcription factors that control the ESR genes [45, 46]. Consistent with this body of work, we observed both the induced and repressed ESR signatures in our environmental perturbation data (Fig 1C). Moreover, unlike most kinases, we observed that inhibition of Tpk1/2/3-as and the TOR-pathway kinases Sch9-as and Ypk1-as had strong transcriptional signatures in the absence of any environmental perturbation and tended to cluster together regardless of environment (Fig 1D and 1E). This result underscores the centrality of these kinases in the stress signaling network.

With the exceptions of the UPR target genes in tunicamycin in Ire1-as cells, and the pheromone responsive genes in NaCl in Hog1-AS and Pbs2-AS cells, we did not identify dedicated sets of target genes regulated by a particular kinase in a specific condition. This is not unexpected, as these connections would have likely already been made in genetic studies. However, as in the case of heat shock, we identified many kinases whose inhibition altered the gene expression response compared to wild type. But rather than modulating the targets of the specific heat shock transcription factor Hsf1 (Fig 2D), these kinase mutants changed the amplitude of the ESR. This is the rule rather than the exception: across the different conditions, different subsets of kinases modulate the ESR. This suggests that the inputs received by Ras/PKA and TOR are diverse and contingent.

The statistical model we employed to analyze the dataset–linear regression–is based on an assumption that the environmental perturbations and kinases each affect the expression of a given gene independently (Fig 4A). When this model fails to explain an experimental measurement, there is a "residual" amount of gene expression that is unaccounted for. In this case, we can infer that the kinase is involved in regulating the gene in the particular environment. By applying this residual analysis globally, we recapitulated known examples of signaling crosstalk and identified putative novel connections (Fig 4). Based on dysregulated genes, we mapped kinases to putative transcription factors (Fig 5). Identifying interactions in this manner was inspired by high-throughput genetic interaction studies [14, 15, 39, 47, 48]. Like these previous works, there are many putative connections that we have not validated. The dataset can be parsed to find these interactions on the accompanying internet application (https://mace.shinyapps.io/kinase-app/).

Finally, we focused on a global antagonistic relationship–previously proposed–that we observed in the data between Tpk1/2/3 and Pbs2, a canonical member of the high osmolarity glycerol (HOG) MAPK pathway [39]. Across all experiments, we found that Pbs2-as cells had

reduced ESR output in both directions (Fig 6B and 6C). Explaining this, we found experimentally that the HOG pathway is a constitutive negative regulator acting upstream of Tpk1/2/3. While we did not reveal the precise molecular mechanism, we traced it to operating at or upstream of Ras-GTP (Fig 6F). These results reveal a new organization to the global stress response signaling network and elaborate the pathway at the center of the ESR.

# Materials and methods

## Yeast strain construction

The 28 kinases genes, including ≥800 bp of promoter/5'UTR sequence were cloned from yeast genomic DNA into single integration vectors and sequence verified. Kinases were rendered analog sensitive (AS) by site-directed mutagenesis by introducing "gatekeeper" mutations as previously described [16, 49]. Yeast strains deleted for the nonessential kinases were transformed with the corresponding AS kinase mutant vector and genomic integration was verified by PCR. Essential kinases were replaced by by first being transformed with an episomal plasmid harboring a *URA3*-marked second copy of the essential kinase, then the genomic copy was deleted, then the AS allele was transformed and verified, and finally the episomal plasmid was counter-selected by 5-FOA resistance. *HSP12pr*-GFP was built by amplifying 800 bp upstream of the *HSP12* ORF from genomic DNA and cloned upstream of the GFP. All strains are in the W303 genetic background. Gene deletions were performed by one-step PCR as described [50]. Site-directed mutagenesis was performed with QuickChange according to the manufacturer's direction (Agilent). Strains used in this work are described in S6 Table.

## Inhibitor cocktail

AS kinase mutants and wild type controls were treated with a cocktail of cell-permeable ATP analogs consisting of 2 μM 1NM-PP1, 2 μM 3MB-PP1 and 2 μM BEZ235 to enable inhibition of kinases with diverse structural features near the ATP binding pocket.

## Cell culture and treatment

Cells were inoculated from single colonies into liquid media (YPD or SDC) and grown for at least 20 hours phase at 30˚C arriving in exponential phase at OD600 ≈ 0.5 at the start of the experiment. Cell growth and treatment were done in deep well 96 well plates. YPD consists of 10 g/L yeast extract, 20 g/L Bactopeptone (Becton Dickinson) and 0.2 g/L dextrose (Sigma-Aldrich). Complete synthetic media (SDC) consists of 6.7 g/L nitrogen base without amino acids, 2 g/L dextrose (Sigma-Aldrich) and 0.79 g/L complete supplement mixture (MP Biomedicals). Cells were then treated with inhibitor cocktail for 5 min, followed by the given environmental perturbation for 20 min. Heat shock was performed by mixing 1 ml of cells at 30˚C with 1 ml of media pre-heated to 50˚C followed by incubation at 39˚C. Glucose depletion was performed by two consecutive whole-plate centrifugation steps (4000g for 3 minutes), decanting off media and replacing with osmo-balanced, glucose-free media (YPSorbitol). All other treatments were applied as 10x stocks made in YPD. Following treatment, cells were then spun (4000g for 3 minutes) in the deep well plates, media decanted and plates were snap frozen in liquid nitrogen. Samples were kept in storage at -80˚C.

## Total RNA extraction

Frozen cell pellets were thawed on ice, resuspended in 1 ml water, transferred to fresh 1.5 ml tubes and harvested by spinning as above. Washed cell pellets were resuspended in 200 μl AE (50 mM NaOAc, pH 5.2, 10 mM EDTA) and vortexed. 20 μl of 10% SDS was added, followed

by 250 µl acid phenol, and samples were incubated at 65˚C with shaking for 10 minutes. After an additional 5 minutes on ice, samples were spun at 15,000 rpm for 5 minutes at 4˚C. Supernatants were transferred to pre-spun heavy phase lock tubes (5 Prime) and 250 µl chloroform was added. Tubes were spun at full speed for 5 minutes at 15,000 rpm and aqueous layers (above the wax) were transferred to fresh 1.5 ml tubes. 30 µl of 3M NaOAc, pH 5.2 was added followed by 750 µL ice cold 100% ethanol. RNA was precipitated at -80˚C for 30 minutes and samples were spun at 15,000 rpm for 30 minutes at 4˚C. Pellets were washed with 1 ml 70% ethanol, spun at 15,000 rpm for 10 minutes at 4˚C. The supernatant was removed and pellets were air dried. Pellets were resuspended in 30 µl DEPC water and the RNA concentrations of the resulting solutions measured by Nano Drop.

## Library preparation and deep sequencing

Sequencing libraries were prepared and sequenced on an Illumina HighSeq 2500 with single end sequencing at 40 base reads. Total RNA samples were submitted to the Whitehead Institute Genome Technology Core where polyA + RNA was purified, fragmented and sequencing libraries barcoded to enable multiplexing in an Illumina Hi-Seq 2500. Reads were assigned by the barcode to the appropriate sample.

## Read alignment and normalization

Genomic alignment was performed using STAR [17], using the UCSC *S. cerevisiae* annotation file. Log counts were calculated using the variance stabilizing transform from DESeq2 [18]. See Computational Pipeline section for more information.

## Differential Gene expression analysis

Differential gene expression calling for each stress-environment were defined using DESeq2 and using an adjusted p-value cutoff of 0.05 using wild type cells with YPD as the baseline condition [18].

## Go enrichment

GO term enrichment was assessed using the Yeastmine API [51].

## Motif enrichment and TF association

Gene's promotor regions were defined as 500bp upstream of the canonical start sequence of the gene. Motif enrichment analysis was conducted with MEME [33], and TF motifs were defined using all the motifs in the JASPAR 2016 Yeast database [34].

## Flow cytometry

Flow cytometry was performed on a BD Fortessa with a high throughput sampler. For each sample, 10000 cells were collected and GFP fluorescence was quantified. The median of the distribution of the GFP signal was determined for three biological replicates, the average and standard deviation of the replicates are presented.

## Supporting information

**S1 Table. Gene ontology terms for clusters.** GO terms and associated p-values for each of the 10 clusters in Fig 1C.
(CSV)

**S2 Table. Differentially expressed genes in each condition.** The set of differentially expressed genes, their relative expression levels and significance values in WT type cells in each condition compared to YPD.
(ZIP)

**S3 Table. Genes putatively regulated by specific kinases in specific conditions.** Genes with residuals that are more than 2.5 standard deviations from the mean in each condition assigned to putative regulatory kinases.
(ZIP)

**S4 Table. Gene ontology analysis of genes putatively regulated by specific kinases in specific conditions.** GO slim terms and associated p-values of sets of genes with residuals that are more than 2.5 standard deviations from the mean in each kinase inhibition/condition dataset.
(CSV)

**S5 Table. Transcription factors linked to kinases in specific conditions.** Transcription factors and associated significance values with motifs enriched among the genes with residuals that are more than 2.5 standard deviations from the mean in each condition and kinase inhibition set.
(CSV)

**S6 Table. Yeast strains used in this study.** Strain IDs and genotypes.
(XLSX)

## Acknowledgments

We would like to acknowledge M. Chevalier, R. Bhatnagar, B. Heineike, and G. Heimberg for valuable discussions. We thank S. Mraz, A. Chilaka, T. Volkert and the Whitehead Institute Genome Technology facility for technical assistance and G. Bell and the Bioinformatics and Research Computing group for guidance and expertise with data analysis.

## Author Contributions

**Conceptualization:** Kieran Mace, Hana El-Samad, David Pincus.

**Formal analysis:** Kieran Mace.

**Funding acquisition:** Hana El-Samad, David Pincus.

**Investigation:** Kieran Mace, Joanna Krakowiak, David Pincus.

**Supervision:** Hana El-Samad, David Pincus.

**Visualization:** Kieran Mace.

**Writing – original draft:** Kieran Mace, Hana El-Samad, David Pincus.

**Writing – review & editing:** Kieran Mace, Hana El-Samad, David Pincus.

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
