## [Decision Letter · Decision Letter 0]

17 Oct 2019

PONE-D-19-26063

Multi-kinase control of environmental stress responsive transcription

PLOS ONE

Dear Dr. David Pincus,

Thank you for submitting your manuscript to PLOS ONE. After careful consideration, we feel that it has merit but does not fully meet PLOS ONE’s publication criteria as it currently stands. Therefore, we invite you to submit a revised version of the manuscript that addresses the points raised during the review process.

As you can see, the reviews are split on your manuscript.

While Reviewer #1 provided a positive response, Reviewer #2 pointed out serious concerns about your study design in its current form.  The reviewers had concerns especially about the lack of strain validation and the lack of biological repeats, which mitigate reliability and significance of the dataset.

After considering all of the points raised by the reviewers, and having looked carefully at the manuscript itself, I have come to the conclusion that we can reconsider your manuscript if you could design further tests which enable to judge the validity of all the downstream analysis.

We would appreciate receiving your revised manuscript by within three months. To enhance the reproducibility of your results, we recommend that if applicable you deposit your laboratory protocols in protocols.io, where a protocol can be assigned its own identifier (DOI) such that it can be cited independently in the future. For instructions see: http://journals.plos.org/plosone/s/submission-guidelines#loc-laboratory-protocols

We look forward to receiving your revised manuscript.

Kind regards,

Reiko Sugiura, M.D., PhD.

Academic Editor

PLOS ONE

Journal Requirements:

1. Thank you for including the following funding information within the acknowledgements section of your manuscript; "This work was supported by NIGMS grant RO1 GM119033 to H.E.-S, NIH Early Independence Award DP5 OD017941 to D.P., and an International Student fellowship from HHMI to K.M"

Reviewers' comments:

Reviewer's Responses to Questions

**Comments to the Author**

1. Is the manuscript technically sound, and do the data support the conclusions?

Reviewer #1: Yes

Reviewer #2: No

2. Has the statistical analysis been performed appropriately and rigorously? 

Reviewer #1: Yes

Reviewer #2: No

3. Have the authors made all data underlying the findings in their manuscript fully available?

Reviewer #1: No

Reviewer #2: Yes

4. Is the manuscript presented in an intelligible fashion and written in standard English?

Reviewer #1: Yes

Reviewer #2: Yes

5. Review Comments to the Author

Reviewer #1: The paper by Mace and coworkers reports a study mostly based on transcriptomic data linking a set of protein kinases and their involvement in the stress response. This is a rather straightforward, sound, and well written study, which provides a huge amount of data and allow reaching interesting conclusions, such as the possible control of the PKA pathway by Pbs2/Hog1 tandem of kinases. I have no major problems with the paper, just a few minor points I would like the authors to consider.

1.- Line 84. The authors select 20 min as a standard time for all stress conditions. Now, the transcriptional response is intrinsically transient, and not all responses to all stresses display the same timing. Why 20 min was selected? I’m not saying this was a wrong choice (actually, I think it makes a lot of sense), but I feel the authors should provide the readers with a short explanation about why 20 min was a reasonable option.

2.- Figure 2B and others. I am not sure the color code is a good option for labeling the 28 different kinases in the figures. There are too many kinases and colors are often repeated (or they are so similar that you cannot tell them apart..), so they are useless. I’d suggest a code based on numbers (i.e. Atg1-1; Cdc15-2, ..and so on). Colors code for stresses would be OK, as they are only 10.

3.- Lines 173-183. These lines are confusing, because authors refer to Fig 3A when they are actually describing Fig 3B (tunicamycin treatment), Ire1 and Cdc15 mutants. The panels need relabeling. In addition, the fact that the order of stress conditions is different for each mutant makes comparisons not easy (this happens in both panels and the same occurs in fig. 6A)

4.- line 184-185. “inhibition of Cdc15 did not affect expression of a specific or dedicated regulon.”. I do have the impression that, in comparison to Ire1, the cdc15 mutant causes specific depletion of the amino acid metabolism GO group. Why the authors do not comment of that?

Reviewer #2: Mace and colleagues have analysed the transcriptomes of 28 S. cerevisae strains each bearing an analogue sensitive mutations in a single kinase and cultivated in 10 different environmental conditions. Based on this dataset they report a series of observation on the role of certain kinases in the yeast response to environmental changes and follow up one observation using a GFP reporter and FACS analysis.

It is a well written paper but I unfortunately have major concerns about the study design.

1) The report provides no evidence that each kinase is actually inactivated after analogue treatment. The authors must have checked their strains and confirmed that the mutations where indeed inactivating the enzymes in their experimental conditions. Why haven’t these data been included? Moreover, in their follow up experiments on figure 6 the authors perform experiments after 1 hour of inactivation and not 5 min. Why is that? The whole study is based on a 5min inactivation time. Without data about strain validation I find it very difficult to judge the validity of all the downstream analysis. What if some or most of the kinases weren't actually inactivated at the time of stress induction?

2) As far as I understand only one repeat of each treatment for each kinase has been performed. With this design I don’t think the authors can provide any statistical support to their analysis. They seem to have used DESeq2 for differential analysis, but I don’t think that using DESeq2 with only one experimental measurement is even statistically valid. Again, given this limitation I am not sure how solid this study conclusions actually are.

3) L84: “We harvested cells following 20 minutes in each environment”. Twenty minutes make sense for oxidative stress but how about glucose depletion for instance. Could the authors justify their choice of time point please?

I am sorry that I cannot be more positive but without strain validation, a sense of the efficieny of kinase inactivation and given the lack of biological repeats, I really struggle to see how much can be learned from this dataset. This being said it would be a shame if these data were not made available to the community as they could help validating other large scale studies. I am not sure however that PLoS ONE is the right home for this.

6. PLOS authors have the option to publish the peer review history of their article (what does this mean?). If published, this will include your full peer review and any attached files.

Reviewer #1: Yes: Joaquin Ariño

Reviewer #2: No

---

## [Author Response · Author response to Decision Letter 0]

19 Feb 2020

We thank the editor and reviewers for their careful reading of the paper and their constructive feedback. Here are our point-by-point responses to the comments.

Please ensure that your manuscript meets PLOS ONE's style requirements, including those for file naming. The PLOS ONE style templates can be found at:

http://www.journals.plos.org/plosone/s/file?id=wjVg/PLOSOne_formatting_sample_main_body.pdf andhttp://www.journals.plos.org/plosone/s/file?id=ba62/PLOSOne_formatting_sample_title_authors_affiliations.pdf

We have reformatted the paper and file names to match the PLOS ONE style.

1. Thank you for including the following funding information within the acknowledgements section of your manuscript; "This work was supported by NIGMS grant RO1 GM119033 to H.E.-S, NIH Early Independence Award DP5 OD017941 to D.P., and an International Student fellowship from HHMI to K.M"

We have removed the funding-related text from the acknowledgements section. Our funding statement should now read: "This work was supported by NIGMS grant RO1 GM119033 to H.E.-S, NIH Early Independence Award DP5 OD017941 to D.P., and an International Student fellowship from HHMI to K.M. The funders had no role in study design, data collection and analysis, decision to publish, or preparation of the manuscript."

Reviewer #1: The paper by Mace and coworkers reports a study mostly based on transcriptomic data linking a set of protein kinases and their involvement in the stress response. This is a rather straightforward, sound, and well written study, which provides a huge amount of data and allow reaching interesting conclusions, such as the possible control of the PKA pathway by Pbs2/Hog1 tandem of kinases. I have no major problems with the paper, just a few minor points I would like the authors to consider.

1.- Line 84. The authors select 20 min as a standard time for all stress conditions. Now, the transcriptional response is intrinsically transient, and not all responses to all stresses display the same timing. Why 20 min was selected? I’m not saying this was a wrong choice (actually, I think it makes a lot of sense), but I feel the authors should provide the readers with a short explanation about why 20 min was a reasonable option.

We have added a justification for this choice of timepoint, writing: “Since stress responses are inherently transient, we chose the 20-minute timepoint to allow for enough time for the stress to be perceived and lead to transcriptional changes, but before the responses were attenuated.”

2.- Figure 2B and others. I am not sure the color code is a good option for labeling the 28 different kinases in the figures. There are too many kinases and colors are often repeated (or they are so similar that you cannot tell them apart..), so they are useless. I’d suggest a code based on numbers (i.e. Atg1-1; Cdc15-2, ..and so on). Colors code for stresses would be OK, as they are only 10.

We agree that providing 28 differentiable colors is incredibly difficult. We attempted implementing the suggestion to use numbering as a second labeling strategy for Kinase strains. However, this was more cluttered and even more difficult to parse. We have minimized the use of the color code and limited it only to the full dataset heatmaps in Figures 1C and 4A, added the key to Figure 4, and made sure to label relevant kinases by name in each panel. 

3.- Lines 173-183. These lines are confusing, because authors refer to Fig 3A when they are actually describing Fig 3B (tunicamycin treatment), Ire1 and Cdc15 mutants. The panels need relabeling. In addition, the fact that the order of stress conditions is different for each mutant makes comparisons not easy (this happens in both panels and the same occurs in fig. 6A)

We have fixed Figure 3 to switch panels A and B to match the text. The ordering of the columns in 3A and 3B are hierarchically generated, grouping kinases of similar effect. We feel that the hierarchical clustering adds additional insights. However, the reviewer is right to point out that the hierarchical clustering in 6A provides very little additional info. We have therefore reordered the two heatmaps in 6A in a consistent way.

4.- line 184-185. “inhibition of Cdc15 did not affect expression of a specific or dedicated regulon.”. I do have the impression that, in comparison to Ire1, the cdc15 mutant causes specific depletion of the amino acid metabolism GO group. Why the authors do not comment of that?

We have now pointed out this observation, saying: “In contrast to Ire1-as, inhibition of Cdc15-as did not exclusively alter any set of genes. However, while not unique to Cdc15-as, inhibition of Cdc15 showed strong repression of the amino acid metabolism genes in both tunicamycin and heat shock (Figure 3A).”

Reviewer #2: Mace and colleagues have analysed the transcriptomes of 28 S. cerevisae strains each bearing an analogue sensitive mutations in a single kinase and cultivated in 10 different environmental conditions. Based on this dataset they report a series of observation on the role of certain kinases in the yeast response to environmental changes and follow up one observation using a GFP reporter and FACS analysis.

It is a well written paper but I unfortunately have major concerns about the study design.

1) The report provides no evidence that each kinase is actually inactivated after analogue treatment. The authors must have checked their strains and confirmed that the mutations where indeed inactivating the enzymes in their experimental conditions. Why haven’t these data been included? Moreover, in their follow up experiments on figure 6 the authors perform experiments after 1 hour of inactivation and not 5 min. Why is that? The whole study is based on a 5min inactivation time. Without data about strain validation I find it very difficult to judge the validity of all the downstream analysis. What if some or most of the kinases weren't actually inactivated at the time of stress induction?

Of the 28 analog sensitive kinases used in this study, 20 have been previously published and validated. We did not develop assays for the 8 remaining kinases to demonstrate that the mutations confer analog-dependent kinase inhibition (Ksp1, Mrk1, Rim11, Rim15, Ssn3, Ste11, Yak1 and Ygk3). We have added this caveat to the text, writing: “For eight of the kinases, the gatekeeper mutant had not been previously generated or validated (Ksp1, Mrk1, Rim11, Rim15, Ssn3, Ste11, Yak1 and Ygk3). We did not develop assays to validate these conditional mutations in this study, so it is possible that the bioinformatically-defined gatekeeper mutations may not confer analog sensitivity to these kinases.”

Five minutes of incubation with the inhibitor is ample time for the concentration of the inhibitor to equilibrate inside and outside of the cell. At the µM concentrations used, binding to the ATP binding pockets (diffusion-limited) should be saturated. Functionally, the evident kinase-dependent gene expression changes we observed – many of which recapitulated previously known biology – demonstrate that 5 minutes was sufficient to inhibit kinase activity. 

The reason we inhibited Tpk1/2/3-as for 1 hour with 1NM-PP1 in the experiments in Figure 6 is because we were assaying the induction of HSP12pr-GFP in response to the inhibitor. This is in contrast to the RNA seq experiments in which we were measuring the global gene expression changes in response to an environmental condition in the context of an inhibited kinase. In order to measure the reporter, we needed to allow time for the transcription, translation and maturation of the GFP to occur prior to measurement. 

2) As far as I understand only one repeat of each treatment for each kinase has been performed. With this design I don’t think the authors can provide any statistical support to their analysis. They seem to have used DESeq2 for differential analysis, but I don’t think that using DESeq2 with only one experimental measurement is even statistically valid. Again, given this limitation I am not sure how solid this study conclusions actually are.

We have four replicates for wild type cells in each of the 10 conditions, and the differentially expressed genes (determined using DESeq2) are defined only in terms of the wild type. We added this caveat to the text: “Due to sequencing constraints, we did not perform experiments in the absence of inhibitor cocktail nor were we able to perform biological replicates for the mutant strains in each condition, limiting our statistical power for any one gene in any mutant in a given condition. However, the four replicates of wild type in each condition provided us with high statistical power for the expression of each gene in the genome in all conditions in the reference strain. While we sequenced RNA from each mutant strain 10 times across the conditions, our choice to sacrifice biological replicates for the mutants in each environment enabled us to broadly survey conditions.”

3) L84: “We harvested cells following 20 minutes in each environment”. Twenty minutes make sense for oxidative stress but how about glucose depletion for instance. Could the authors justify their choice of time point please?

We have added a justification for this choice of timepoint, writing: “Since stress responses are inherently transient, we chose the 20-minute timepoint to allow for enough time for the stress to be perceived and lead to transcriptional changes, but before the responses were attenuated.” With respect to glucose depletion specifically, in fact glucose depletion resulted in the largest magnitude of gene expression changes in the dataset (see Fig. 1C), indicating that the 20-minute timepoint enabled us to monitor a cellular response.

I am sorry that I cannot be more positive but without strain validation, a sense of the efficieny of kinase inactivation and given the lack of biological repeats, I really struggle to see how much can be learned from this dataset. This being said it would be a shame if these data were not made available to the community as they could help validating other large scale studies. I am not sure however that PLoS ONE is the right home for this.

We hope that our responses have provided rationale our experimental design and that the caveats added to the text describe the limitations of the dataset. The fact that we recapitulated so much known biology (e.g., crosstalk between osmo-sensing and pheromone signaling, the role of Ire1 in the ER stress response) should also be reassuring that the data are relevant and warrant publication in PLoS One.

We have performed the PACE diagnostics of our figures.

---

## [Decision Letter · Decision Letter 1]

26 Feb 2020

Multi-kinase control of environmental stress responsive transcription

PONE-D-19-26063R1

Dear Dr. David Pincus,

We are pleased to inform you that your manuscript has been judged scientifically suitable for publication and will be formally accepted for publication once it complies with all outstanding technical requirements.

With kind regards,

Reiko Sugiura, M.D., PhD.

Academic Editor

PLOS ONE

Additional Editor Comments (optional):

Reviewers' comments:

Reviewer's Responses to Questions

**Comments to the Author**

1. If the authors have adequately addressed your comments raised in a previous round of review and you feel that this manuscript is now acceptable for publication, you may indicate that here to bypass the “Comments to the Author” section, enter your conflict of interest statement in the “Confidential to Editor” section, and submit your "Accept" recommendation.

Reviewer #1: All comments have been addressed

Reviewer #2: (No Response)

2. Is the manuscript technically sound, and do the data support the conclusions?

Reviewer #1: (No Response)

Reviewer #2: Yes

3. Has the statistical analysis been performed appropriately and rigorously? 

Reviewer #1: (No Response)

Reviewer #2: Yes

4. Have the authors made all data underlying the findings in their manuscript fully available?

Reviewer #1: (No Response)

Reviewer #2: Yes

5. Is the manuscript presented in an intelligible fashion and written in standard English?

Reviewer #1: (No Response)

Reviewer #2: Yes

6. Review Comments to the Author

Reviewer #1: (No Response)

Reviewer #2: I thank the authors for their replies. I am now reassured that most strains have been validated and understand the choice of experimental conditions better. In addition, given that the limitations of the design have now been clearly stated, I see no reason to delay the publication of this study further.

7. PLOS authors have the option to publish the peer review history of their article (what does this mean?). If published, this will include your full peer review and any attached files.

Reviewer #1: Yes: Joaquín Arino

Reviewer #2: No

---

## [Editor Report · Acceptance letter]

2 Mar 2020

PONE-D-19-26063R1 

Multi-kinase control of environmental stress responsive transcription 

Dear Dr. Pincus:

I am pleased to inform you that your manuscript has been deemed suitable for publication in PLOS ONE. Congratulations! Your manuscript is now with our production department. 

With kind regards,

on behalf of

Dr. Reiko Sugiura 

Academic Editor

PLOS ONE